# Malnutrition disrupts adaptive immunity during visceral leishmaniasis by enhancing IL-10 production

**Laís Amorim Sacramento, Claudia Gonzalez-Lombana, Phillip Scott**ⓘ*

Department of Pathobiology, School of Veterinary Medicine, University of Pennsylvania, Pennsylvania, United States of America

* pscott@vet.upenn.edu

## Abstract

Protein-energy malnutrition (PEM) is a risk factor for developing visceral leishmaniasis (VL). While nutrient deficiency can impair immunity, its mechanistic impact on protective adaptive immune responses following *Leishmania* infection remains unknown. To determine the potential negative impacts of malnutrition on anti-parasitic responses in chronic VL, we provided mice with a polynutrient-deficient diet (deficient protein, energy, zinc, and iron) that mimics moderate human malnutrition. The polynutrient-deficient diet resulted in growth stunting and reduced mass of visceral organs and following infection with *Leishmania infantum*, malnourished-mice harbored more parasites in the spleen and liver. Malnourished and infected mice also had fewer T lymphocytes, with reduced T cell production of IFN-γ required for parasite clearance and enhanced production of the immunosuppressive cytokine, IL-10. To determine if IL-10 was causative in disease progression in the malnourished mice, we treated infected mice with monoclonal antibody α-IL-10R. α-IL-10R treatment reduced the parasite number in malnourished mice, restored the number of T cells producing IFN-γ, and enhanced hepatic granuloma formation. Our results indicate that malnutrition increases VL susceptibility due to defective IFN-γ-mediated immunity attributable to increased IL-10 production.

**Data Availability Statement:** All data are in the manuscript and Supporting information files.

## Author summary

Malnutrition contributes to the development of visceral leishmaniasis (VL) following *Leishmania* infection. Despite the clear association, the impact of malnutrition on the adaptive immune mechanisms required for parasite control is still unclear. We found that malnutrition disrupts the ability to control parasite replication in the spleen and liver due to defective IFN-γ-mediated immunity, reduced hepatic granuloma formation, and enhanced IL-10 production. Blocking IL-10R signaling restored the protective mechanisms to control parasite replication in the malnourished mice without interfering with the undernutrition state. Thus, we demonstrate that increased IL-10 production driven by malnutrition disrupts protective immunity against *Leishmania*, thereby enhancing the

**Funding:** This work was supported by National Institutes of Health grant RO1-AI150606 (PS). The funder had no role in study design, data collection and analysis, decision to publish, or preparation of the manuscript.

**Competing interests:** The authors have declared that no competing interests exist.

susceptibility of VL. Understanding the association between malnutrition and VL will provide insights into therapeutic approaches to treat this aggressive disease.

## Introduction

Protein-energy malnutrition (PEM) is the most frequent type of malnutrition, affecting at least 800 million people worldwide, especially children and elderly individuals [1], and is the most common cause of immunodeficiency worldwide [2]. Deficiency in macronutrients and/or micronutrients induces impaired immunity and consequently increases susceptibility to various pathogens [3–5]. The risk of developing visceral leishmaniasis (VL) following infection with *Leishmania* parasites is increased by malnutrition [6–8]. Moreover, murine models of moderate PEM demonstrate that malnutrition causes early dissemination of *Leishmania* parasites to visceral organs due to defective innate immunity [9–12]. Despite substantial efforts to define the mechanisms between malnutrition and VL, the immunopathogenesis of this association is not completely understood. This study aimed to determine the impact of malnutrition on adaptive immunity during VL. We took advantage of a well-established model of polynutrient deficiency that mimics moderate human malnutrition [13] to fill this knowledge gap. We found that malnutrition increases the susceptibility to VL due to defective IFN-γ-mediated immunity and hepatic granuloma formation due to an aberrant IL-10 response.

## Results

### Malnutrition disrupts anti-parasitic immunity during visceral leishmaniasis

We first characterized the impact of the polynutrient-deficient diet (PND diet) on the nutritional status of our model by monitoring body weight, tail lengths, and the size of the spleen and liver, which are the target organs of VL. Mice fed the PND diet gained significantly less weight over time than mice fed the control diet, but there was no statistical difference between naïve and infected malnourished mice (Fig 1A and 1B). Consistent with previous reports, the growth curve observed in PND diet-fed mice was considered a moderate level of malnutrition in mice that mimics malnutrition in humans [13]. At the end of 6 weeks of feeding, mice on the PND diet had shorter tails (Fig 1C) and there was a significant reduction in the weight of the spleen (Fig 1D) and liver (Fig 1E). These data suggest that a PND diet led to growth stunting and reduced visceral organ size.

To test whether malnutrition impacts the control of parasite replication, we performed limiting dilution to quantify parasite number in the target organs of VL. Infected-malnourished mice harbored significantly more parasites at the spleen and liver at 4 weeks post-infection (wpi) (Fig 1F and 1G). An assessment of immune cells revealed a reduction in the number of CD3$^+$ cells (Fig 1H) and CD4$^+$ T cells (Fig 1I) in the spleen of infected-malnourished mice compared to infected-control. No significant differences were observed between non-infected malnourished versus control mice. Moreover, the development of Th1-type immunity critical for parasite control [14,15] appears to be compromised, as we found a reduced number of CD4$^+$ T-bet$^+$ T cells in the spleen of infected-malnourished mice compared with infected-control mice (Fig 1J). Consistently, the number of CD4$^+$ T cells producing IFN-γ was significantly reduced in the spleen of infected-malnourished mice compared with infected-control mice (Fig 1K).

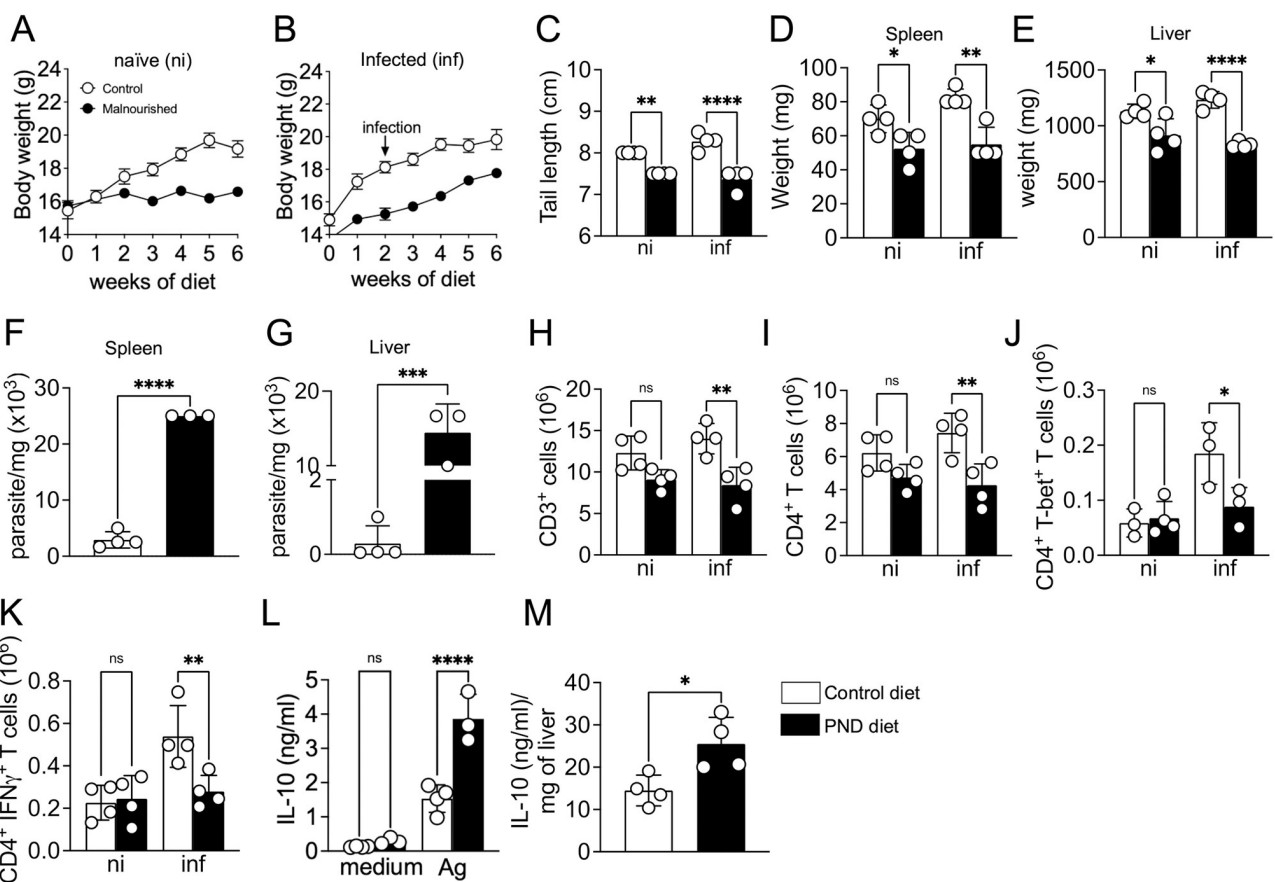

**Fig 1. Malnutrition disrupts anti-parasitic immunity during visceral leishmaniasis.** (A and B) Body weight of naive or *L. infantum*-infected control diet or polynutrient deficient diet (PND diet) over time. (C) Final tail lengths at 6 weeks of diet and 4 weeks post-infection (wpi). (D) Spleen and (E) liver weight. Parasite number in the (F) spleen and (G) liver of control diet and PND diet-fed mice at 4 weeks post-infection. Absolute number of (H) CD3+ cells, (I) CD4+ T cells, (J) CD4+ T cells expressing T-bet, and (K) CD4+ T cells expressing IFNγ in the spleen of control and PND diet-fed mice. Cells were gated based on their characteristic size (FSC) and granularity (SSC), singlet cells, live+, and CD45+ cells. (L) IL-10 levels in the supernatants from splenocytes restimulated with *L. infantum* crude antigen or medium for 72 h at 4 wpi. (M) IL-10 levels in the liver homogenate supernatant at 4 wpi. The data are expressed as mean ± SEM and represent two experiments (n = 4 mice/group). The statistical significance was calculated by one-way ANOVA or Student's t-test (*p < 0.05, **p < 0.01, ***p < 0.001 and ****p < 0.0001).

IL-10, a key immunosuppressive factor, contributes to the pathology of VL [16–20]. Notably, IL-10 levels were significantly higher in the supernatant of cultured cells from infected-malnourished mice versus the infected-control mice (Fig 1L). Consistently, liver samples from infected-malnourished mice exhibited increased amounts of IL-10 (Fig 1M). The flow cytometry analysis demonstrates that IL-10 is produced by many cells during VL, especially by Th1 cells (Tr1), as previously reported [21,22]. Interestingly, malnutrition enhances the frequency of NK cells and monocytes producing IL-10 (S1 Fig). These data suggest that malnutrition results in increased IL-10 production, thereby compromising the development of a protective Th1 response and a consequent increase in parasite burden.

## IL-10R blockade restores anti-parasitic immunity

IL-10 is a pleiotropic cytokine that can influence VL development through multiple mechanisms [18–21]. To investigate the effect of IL-10 production in the infected malnourished mice, we treated mice with either α-IL-10R mAb to block IL-10 signaling or an α-IgG mAb as

a control. Treatment was initiated at the time of infection and continued weekly. The treatment with α-IL-10R did not alter the body weight (Fig 2A and 2B) or tail length (Fig 2C) of malnourished mice. We observed a small increase in the spleen (Fig 2D) and liver (Fig 2E) weight of PND diet-fed mice treated with α-IL-10R compared with the control α-IgG mAb.

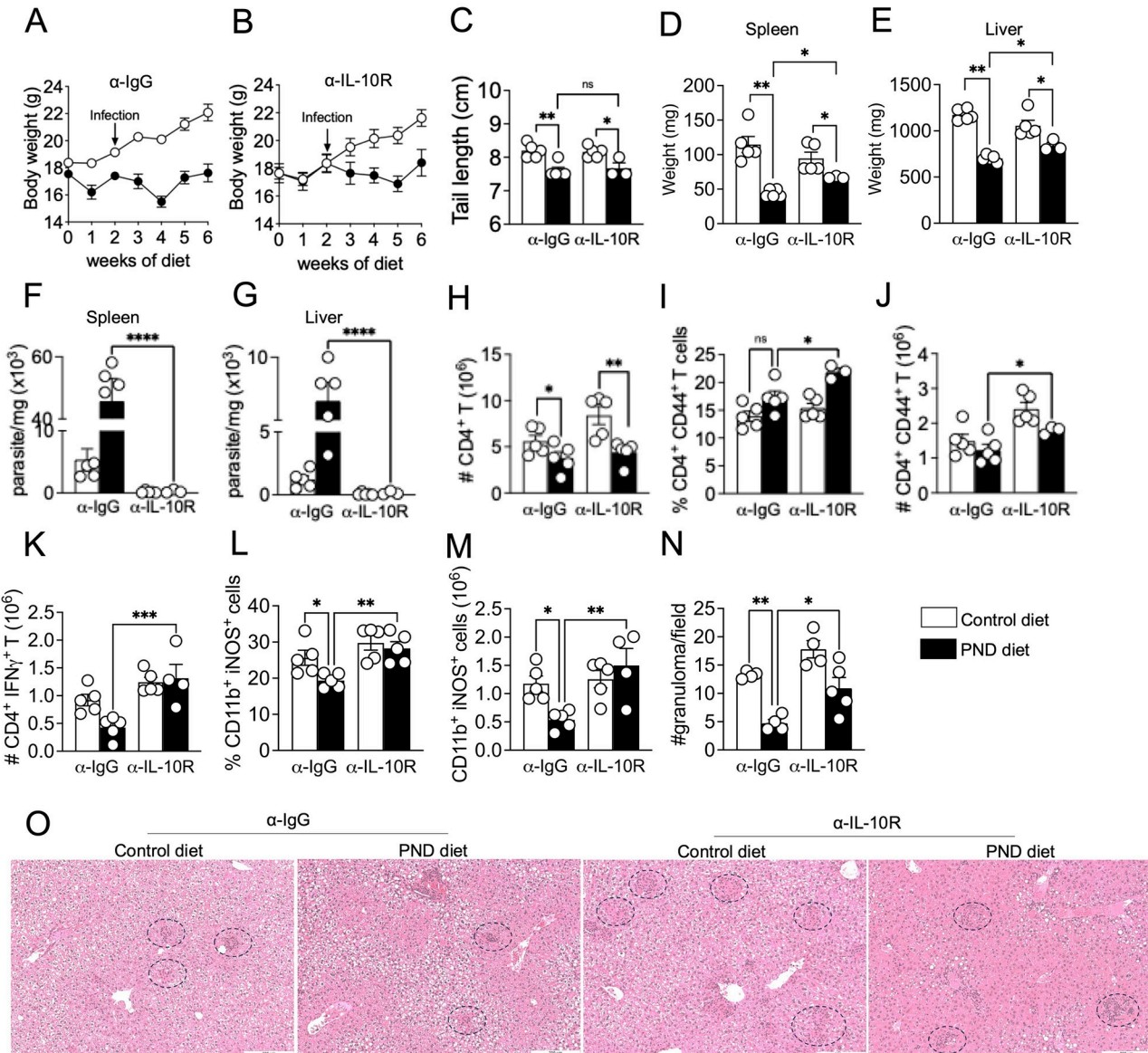

**Fig 2. IL-10R blockade restores anti-parasitic immunity.** *L. infantum*-infected mice fed with control diet or PND diet were treated with anti-IgG control or anti-IL-10R antibody and euthanized at 4 wpi. Treatment was initiated at the time of infection and continued weekly until the termination of the experiment. (A and B) Body weight of infected mice treated with (A) α-IgG control or (B) α-IL-10R antibody. (C) Tail lengths at 6 weeks of diet and 4 weeks post-infection (wpi). (D) Spleen and (E) liver weight. Parasite number in the (F) spleen and (G) liver. Absolute number of (H) CD4+ T cells, (I) percentage and (J) number of CD4+ T cells expressing CD44, (K) number of CD4+ T cells expressing IFNγ at the spleen of control and PND diet-fed mice treated with anti-IgG control or anti-IL-10R antibody. (L-M) Percentage and number of CD11b+ cells expressing iNOS at the spleen of infected control and PND diet-fed mice treated with α-IgG control or α-IL-10R antibody. Cells were gated based on their characteristic size (FSC) and granularity (SSC), singlet cells, live+, and CD45+ cells. T cells were gated on CD3+, CD4+, and myeloid cells on CD11b+, CD11b+ iNOS+ cells. (N-O) Representative pictures of H&E-stained liver sections (20x magnification) and bar graph of the number of granulomas per field (counted in 10x magnification). The data are expressed as mean ± SEM and represent two experiments (n = 3–5 mice/group). The statistical significance was calculated by one-way ANOVA (*p < 0.05, **p < 0.01, ***p < 0.001 and ****p < 0.0001).

As previously reported, α-IL-10R treatment reduced the parasite number in the spleen and liver of well-nourished mice [21]. Consistently, IL-10 signaling blockade also reduced the parasite burden in the target organs of infected-malnourished mice (Fig 2F and 2G). However, while we did not observe a difference in CD4$^+$ T cell numbers in the spleen of α-IL-10R mAb treated mice compared with α-IgG control mice (Fig 2H), we did find that α-IL-10R treatment increased the percentage and number of CD4$^+$ CD44$^+$ T cells (Fig 2I and 2J) and the number of CD4$^+$ IFNγ$^+$ T cells in the spleen of the infected-malnourished mice (Fig 2K). Thus, our data support a mechanistic model in which the enhanced IL-10 production in infected-malnourished mice prevents the generation of CD4$^+$ IFNγ$^+$ T cells required for protection.

In addition to a protective Th1 response, iNOS-mediated nitric oxide production is essential for controlling parasites within macrophages [18–20]. Therefore, we next evaluated whether IL-10R blockade interferes with the expression of iNOS in malnourished infected mice. Notably, treatment with α-IL-10R mAb enhanced the percentage and number of CD11b$^+$ cells expressing iNOS from the spleen of infected malnourished mice (Fig 2L and 2M), supporting a pleiotropic role for this immunosuppressive cytokine.

Finally, control of liver infection during experimental VL depends on granuloma formation [23]. We, therefore, evaluated the liver sections to determine whether the enhanced IL-10 production in the infected-malnourished mice interfered with hepatic granuloma generation. α-IL-10R mAb treatment increased the granuloma formation in the control mice, as previously reported [19]. Notably, infected-malnourished mice had a reduced number of hepatic granulomas compared to control mice. Importantly, the blockade of IL-10 signaling partially restored the number of granulomas in the liver of infected malnourished mice (Fig 2O). We measured cytokine levels in liver samples to investigate whether IL-10R blockade affects systemic inflammatory markers. Corroborating the previous report, malnourished mice have increased levels of IL-1β, IL-6, and TNF-α compared with control mice at baseline [24], and the α-IL-10R mAb treatment has no impact on the modulation of systemic cytokine levels (S2 Fig). Our results indicate that malnutrition increases VL susceptibility due to defective activation and expansion of CD4$^+$ IFNγ$^+$ T cells, decreased iNOS expression, and reduced hepatic granuloma formation, which is attributable to increased IL-10 production.

## Discussion

Malnutrition disrupts innate immune mechanisms, causes early visceralization of parasites during VL [9,11,13], and can affect T cell-mediated responses and granuloma formation [25–27]. While increased IL-10 expression is also associated with VL progression in humans [17], the potential connection between these observations has remained unclear. Here, we found that malnutrition increases VL susceptibility due to an elevation in IL-10 production, which limits IFNγ-mediated immunity, iNOS production, and hepatic granuloma formation. Moreover, we found that α-IL-10R treatment restored activation and expansion of T cells producing IFNγ in the malnourished-infected mice without altering the undernutrition observed. The restoration of iNOS expression following α-IL-10R treatment could be either a consequence of enhanced CD4$^+$ IFNγ$^+$ T cells or a direct effect on the myeloid cells [17,28].

In agreement with our data, high levels of systemic IL-10 have been reported in malnourished mice challenged intraperitoneally with bacterial LPS [29–31]. The observed increase in circulating levels of inflammatory cytokines in malnourished individuals [32,33] and murine models of malnutrition [24] has been attributed to a dysfunctional intestinal barrier and bacterial translocation [24]. Based on these findings, we hypothesize that the increased IL-10 production observed in the malnourished model results from an excessive systemic inflammatory process. In sepsis, high levels of IL-10 are critical for controlling inflammatory responses that

can cause tissue damage [34,35]. In support, VL patients have a systemic overproduction of inflammatory mediators [36,37], and the high levels of IL-10 limit tissue damage in this disease. However, the tissue-preserving capacity of IL-10 is counterbalanced by its ability to generate a permissive environment for parasite replication [38–40][16]. Additionally, malnutrition disturbs circulating metabolite levels, enhancing tryptophan catabolites such as kynurenine [41]. Given the influence of tryptophan catabolism on immunosuppression, the high level of IL-10 might result from kynurenine axis stimulation [42,43]. Specifically, our findings indicate that the enhanced IL-10 levels in malnutrition further contribute to VL by disrupting Th1 immunity and hepatic granuloma formation, thereby limiting parasite clearance. Further mechanistic dissection of this pathway may reveal opportunities to enhance the protective immune response without exacerbating tissue pathology.

## Material and methods

### Ethics statement

This study followed the Guide for the Care and Use of Laboratory Animals (National Institutes of Health). The protocol was approved by the Institutional Animal Care and Use Committee, University of Pennsylvania Animal Welfare Assurance Number 806959.

### Mice and diets

4-week-old female C57BL/6 mice were purchased from Charles River. Mice were grouped in cages of 4–5 and were maintained in a specific pathogen–free facility with free access to water, nesting material, and housed at a temperature of 21˚C at the University of Pennsylvania Animal Care Facilities. Mice received a control standard chow (17% protein, 100 ppm Iron, 30 ppm zinc—Teklad diet, #99103) or an isocaloric but protein, iron, and zinc-deficient diet (3% protein, 10 ppm Iron, 1 ppm zinc—Teklad diet, #99075). After 2 weeks of diet, mice were divided into two subgroups, one of which was infected with *L. infantum*. Polynutrient deficient diet-fed (PND diet) mice received 90% of the consumed food of the control mice over weeks calculated based on the food ingestion (g/mouse/day) of the control diet group obtained every 72 hours. The body weight of mice was monitored once a week. All animals were used in accordance with the recommendations in the Guide for the Care and Use of Laboratory Animals of the National Institutes of Health and the guidelines of the University of Pennsylvania Institutional Animal Use and Care Committee. The protocol was approved by the Institutional Animal Care and Use Committee, University of Pennsylvania Animal Welfare Assurance.

### Parasite culture and infection

*L. infantum* (LLM-320) was cultured in Schneider medium with 20% heat-inactivated fetal bovine serum, 5% penicillin, and streptomycin. The mice were intravenously infected via the retro-orbital plexus with $10^7$ *L. infantum* metacyclic enriched promastigotes in 100 μL of PBS. A quantitative limiting dilution assay determined parasite burdens.

### In vivo α-IL-10R antibody treatment

Mice were treated intraperitoneally twice a week with 500 μg of α-mouse IL-10R antibody or rat anti-IgG1 isotype control. Treatment was initiated at the time of infection and continued weekly until the termination of the experiment at 4 weeks post-infection.

### Statistical analysis

Data are shown as means ± SEM. Statistical significance was determined using the two-tailed unpaired Student's t-test or one-way ANOVA. All statistical analysis was calculated using GraphPad Prism Software. Differences were considered significant when $^*p < 0.05$, $^{**}p \le 0.01$, $^{***}p \le 0.001$, $^{****}p < .0001$.

## Supporting information

**S1 Fig. IL-10 production by lymphoid and myeloid populations in malnourished mice.** The percentage of IL-10 in cells from the spleen and liver of naive or *L. infantum*-infected control diet or polynutrient deficient diet (PND diet) at 6 weeks of diet and 4 weeks post-infection. Leukocytes from the spleen and liver were stimulated with PMA, ionomycin, and monensin in the presence of BFA for 2.5 hr and subsequently harvested and stained for IL-10. Cells were gated based on size (FSC) and granularity (SSC), singlet cells, live$^+$, and CD45$^+$ cells. For NK and NKT cell identification, the NK1.1$^+$ subset was identified as CD3$^-$NK1.1$^+$ and CD3$^+$NK1.1$^+$, respectively. NK1.1$^-$ subset was gated on CD3$^+$ subset for subsequent identification of CD4$^+$IFNγ$^+$, CD4$^+$ Foxp3$^+$, and CD8$^+$ cells. CD3$^-$ subset was followed by gate on CD19$^+$ subset for B cell. For myeloid population identification, the CD19$^-$ subset was followed by the gate on CD11c$^+$Ly6C$^-$ (dendritic cells), CD11c$^-$Ly6C$^+$CD11b$^+$ (monocytes), and CD11c$^-$Ly6C$^-$F4/80$^+$ (macrophages). The statistical significance was calculated by one-way ANOVA ($^*p < 0.05$, $^{**}p < 0.01$, and $^{***}p < 0.001$). (DOCX)

**S2 Fig. IL-10R blockade has no impact on systemic inflammation.** Levels of IL-1β, IL-6, and TNFα in supernatants from a liver homogenate at 6 weeks of diet and 4 weeks post-infection. The data are expressed as mean ± SEM and represent the combination of two experiments. The statistical significance was calculated by one-way ANOVA ($^*p < 0.05$, $^{**}p < 0.01$, and $^{***}p < 0.001$). (DOCX)

**S1 Supporting_Protocol. Detailed description of methods.** (DOCX)

## Acknowledgments

We thank the Comparative Pathology Core (CPC).

## Author Contributions

**Conceptualization:** Laís Amorim Sacramento, Phillip Scott.

**Data curation:** Phillip Scott.

**Formal analysis:** Laís Amorim Sacramento.

**Funding acquisition:** Phillip Scott.

**Investigation:** Laís Amorim Sacramento, Phillip Scott.

**Methodology:** Laís Amorim Sacramento, Claudia Gonzalez-Lombana, Phillip Scott.

**Project administration:** Phillip Scott.

**Resources:** Phillip Scott.

**Supervision:** Phillip Scott.

**Validation:** Laís Amorim Sacramento, Phillip Scott.

**Visualization:** Laís Amorim Sacramento, Phillip Scott.

**Writing – original draft:** Laís Amorim Sacramento.

**Writing – review & editing:** Laís Amorim Sacramento, Phillip Scott.

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
