## [Decision Letter · Decision Letter 0]

5 Aug 2024

Dear Prof. Scott,

Thank you very much for submitting your manuscript "Malnutrition disrupts adaptive immunity during visceral leishmaniasis by enhancing IL-10 production" for consideration at PLOS Pathogens. As with all papers reviewed by the journal, your manuscript was reviewed by members of the editorial board and by several independent reviewers. In light of the reviews (below this email), we would like to invite the resubmission of a significantly-revised version that takes into account the reviewers' comments.

Your studies were deemed to be interesting. However, both reviewers commented on the absence of mechanistic insight into your observations. Please pay heed to the suggestions from Reviewer 2 for guidance on experiments that would provide mechanistic insight.

We cannot make any decision about publication until we have seen the revised manuscript and your response to the reviewers' comments. Your revised manuscript is also likely to be sent to reviewers for further evaluation.

Sincerely,

Peter E Kima

Guest Editor

PLOS Pathogens

Margaret Phillips

Section Editor

PLOS Pathogens

Michael Malim

Editor-in-Chief

PLOS Pathogens

orcid.org/0000-0002-7699-2064

Your studies were deemed to be interesting. However, both reviewers commented on the absence of mechanistic insight into your observations. Please pay heed to the suggestions from Reviewer 2 for guidance on experiments that would provide mechanistic insight.

Reviewer's Responses to Questions

**Part I - Summary**

Reviewer #1: Sacramento et al., use a poly-nutrient deficient mouse model that mimics the most commonly observed malnutrition to study the immune mechanism of L. infantum infection mediated pathogenesis. Previous studies using the PND mouse models reported deficient innate immune responses, but the impact on adaptive immunity remained unexplored. In this study the authors report elevated IL-10 levels under the PND regimen and that blocking IL-10 controlled the splenic parasite burden. Overall the studies are well conducted and the conclusions are robust.

Reviewer #2: The manuscript by Sacramento, et al, describes the outcome of chronic systemic Leishmania infantum infection of malnourished vs. control mice. It identifies increased IL-10 production as a driver of the increased severity of L. infantum infection in malnourished mice because antibody-mediated blockade of the IL-10 receptor improved IFNg production by CD4 T cells, hepatic granuloma formation, and parasite control. The experimental design and execution are excellent and the manuscript is well-written. While the results are interesting and add to our understanding of the impact of malnutrition on adaptive anti-Leishmania immunity, the data stop short of establishing a clear mechanistic understanding of the contribution of malnutrition to the immunopathogenesis of visceral leishmaniasis.

**Part II – Major Issues: Key Experiments Required for Acceptance**

Reviewer #1: (No Response)

Reviewer #2: 1. The source of IL-10 is critical to understanding how malnutrition influences the immunopathogenesis of visceral leishmaniasis. Is it driven by polarized or mixed-phenotype CD4 T cells or part of a pathologic innate immune response as has been previously reported in this model? It would be relatively straightforward to determine the cellular source of the IL-10 (CD4 T cells or innate immune cells) by intracellular staining as was done for IFNg and iNOS.

2. Since myeloid cells expressing iNOS were reduced in PND mice, it would be helpful to determine if this was a direct autocrine/paracrine effect of IL-10, impaired IFNg responsiveness of myeloid cells in PND, or indirect through suppression of T cell IFNg production. This could be teased out with ex vivo cell culture experiments.

3. Since exaggerated systemic inflammation in this model seems to be the driver of growth faltering (reference 30), and parasite dissemination from the skin (doi.org/10.1371/journal.pntd.0011040), and the increased IL-10 could be an innate regulatory response to control the inflammation, it is surprising that IL-10R blockade did not worsen the growth metrics. Perhaps this supports the notion that IL-10R blockade is primarily targeting the T cell anti-Leishmania adaptive response. As noted above, ex vivo and in vitro studies could help sort this out. Although it is not the focus of this manuscript, it would be interesting to know how IL-10R blockade affected systemic inflammatory markers.

**Part III – Minor Issues: Editorial and Data Presentation Modifications**

Reviewer #1: The data presented in the manuscript supports the hypothesis that the main mechanism of VL pathogenesis in a malnutrition condition to be IL-10 mediated. Blocking of IL-10 did not alter the number of CD4+ T cells but improved the IFN-g producing CD4+ T cells along with iNOS+ cells. Accordingly, liver and spleen parasite burdens were controlled upon IL-10 blocking treatment of L. infantum infected PND mice.

While the manuscript shows the correlation between malnutrition and an IL-10 mediated anti-inflammatory response, which was also observed in sepsis and fungal infections, a mechanistic understanding is currently lacking. Several bacterial/fungal infection studies reported the identification of IDO1-AhR signaling mediated IL-10 production. The authors may expand the discussion section on potential metabolic immune regulation that may underlie the elevated IL-10 expression. Please see https://doi.org/10.1007/s00394-018-1679-0 for an excellent coverage.

Have the authors considered a rescue group where normal diet is provided to the PND mice after the L infantum infection to verify if IL-10 levels alter in sync with the nutrition?

Reviewer #2: 4. The authors should recognize the work of the Cuervo group (DOI:10.1371/journal.pone.0114584) who previously showed increased parasite load, increased IL-10 and reduced IFNg production in the spleen of L. infantum infected protein-deficient mice.

5. It would be helpful to label the tissue sources in the different panels in Figs. 1 and 2.

6. Reference 11 does not seem to fit with where it is cited.

PLOS authors have the option to publish the peer review history of their article (what does this mean?). If published, this will include your full peer review and any attached files.

Reviewer #1: **Yes: **Sreenivas Gannavaram

Reviewer #2: No
---

## [Editor Report · Decision Letter 1]

2 Nov 2024

Dear Prof. Scott,

We are pleased to inform you that your manuscript 'Malnutrition disrupts adaptive immunity during visceral leishmaniasis by enhancing IL-10 production' has been provisionally accepted for publication in PLOS Pathogens.

Best regards,

Peter E Kima

Guest Editor

PLOS Pathogens

Margaret Phillips

Section Editor

PLOS Pathogens

Michael Malim

Editor-in-Chief

PLOS Pathogens

orcid.org/0000-0002-7699-2064
---

## [Editor Report · Acceptance letter]

6 Nov 2024

Dear Prof. Scott,

We are delighted to inform you that your manuscript, "Malnutrition disrupts adaptive immunity during visceral leishmaniasis by enhancing IL-10 production," has been formally accepted for publication in PLOS Pathogens.

Best regards,

Michael Malim

Editor-in-Chief

PLOS Pathogens

orcid.org/0000-0002-7699-2064